# Wave Packet Dynamical Simulation of Quasiparticle Interferences in 2D Materials †

Péter Vancsó [1,*], Alexandre Mayer [2], Péter Nemes-Incze [1] and Géza István Márk [1]

1    Centre for Energy Research, Institute of Technical Physics and Materials Science, 1121 Budapest, Hungary; nemes.incze.peter@ek-cer.hu (P.N.-I.); mark@mfa.kfki.hu (G.I.M.)
2    Department of Physics, University of Namur, 5000 Namur, Belgium; alexandre.mayer@unamur.be
*    Correspondence: vancso.peter@energia.mta.hu
†    This paper is an extended version of paper published in the 1st International Electronic Conference on Applied Sciences, 10–30 November 2020.

**Abstract:** Materials consisting of single- or a few atomic layers have extraordinary physical properties, which are influenced by the structural defects. We present two calculation methods based on wave packet (WP) dynamics, where we compute the scattering of quasiparticle WPs on localized defects. The methods are tested on a graphene sheet: (1) We describe the perfect crystal lattice and the electronic structure by a local atomic pseudopotential, then calculate the Bloch eigenstates and build a localized WP from these states. The defect is represented by a local potential, then we compute the scattering by the time development of the WP. (2) We describe the perfect crystal entirely by the kinetic energy operator, then we calculate the scattering on the local defect described by the potential energy operator. The kinetic energy operator is derived from the dispersion relation, which can be obtained from any electronic structure calculation. We also verify the method by calculating Fourier transform images and comparing them with experimental FFT-LDOS images from STM measurements. These calculation methods make it possible to study the quasiparticle interferences, inter- and intra-valley scattering, anisotropic scattering, etc., caused by defect sites for any 2D material.

**Keywords:** wave packet dynamics; quasiparticle interference; graphene; transport properties; defect





## 1. Introduction

To design nanoelectronic devices from 2D materials, it is important to precisely understand the dynamics of electrons in these structures. While Bloch waves propagate freely in the perfect crystal, in the presence of defects [1–3] the quasi-particles will be scattered and interfered with themselves, which will influence the transport properties of the material [4–6]. The interference of incoming and scattered waves leads to characteristic patterns in the local density of states (LDOS) around the defects, called quasiparticle interference (QPI) patterns [7,8] also referred to as Friedel oscillations [9].

The investigation of the QPI patterns around defects not only gives information on the type of the impurity, but also on more hidden band structure properties of the host material. A prime example of graphene, where the chiral property of the Dirac electrons has a substantial effect on the scattering processes [10]. In graphene two main types of scattering processes are possible: intravalley scattering within one $K$ ($K'$) valley or intervalley scattering between two neighboring $K$ and $K'$ valleys. Due to the honeycomb structure of the graphene lattice, a new degree of freedom, the pseudospin, emerges. It was proved earlier that the conservation of pseudospin during the scattering process leads to suppression of intravalley backscattering resulting in a topological fingerprint in the QPI patterns.

Experimentally, the real-space modulation of the LDOS is accessible by fast Fourier transformation (FFT) of the topography or tunneling conductance maps, measured by scanning tunneling microscopy [11]. The great advantage of the STM is that it allows

us to understand the basic mechanism of quasiparticle scattering at the atomic scale. By using this method, the quasiparticle chirality effects were observed in graphene [12–14]. However, the experimentally observed QPI often has complicated patterns because the elastic scattering mixes states that are located on the same quasiparticle contour of constant energy (CCE). If the material has an intricate Fermi surface, where multiple scattering effects can take place, the measured FFT-STS images display more complex characters and theoretical investigations are highly necessary to understand the experiments [15,16].

Wave packet dynamics (WPD) [17] can simulate electronic dynamics including multiple scattering processes at the nanoscale and are capable of calculating realistic models containing several hundred atoms already on a personal computer. The physical system is described by a Hamiltonian and the initial conditions are given by an initial wave function. Solution of the time-dependent Schrödinger equation then yields the $\psi\left(\vec{r}, t\right)$ time-dependent wave function and its time-energy Fourier transform gives the $\psi\left(\vec{r}, E\right)$ energy-dependent wave function, which can be used for the interpretation of the QPI patterns measured by STM. In the one particle approximation, we calculate only a one (quasi)particle three-dimensional (3D) wave function instead of the 3N dimensional many-body wave function and the details of the many-body interactions are coded into the Hamilton operator. This can be achieved by building an appropriate $V\left(\vec{r}\right)$ one-particle pseudopotential. During the past decade we were performing WPD calculations [18–20] for many $sp^2$ carbon nanosystems by using a variationally calculated local carbon one-electron pseudopotential [21]. This pseudopotential has two major advantages: (i) it brings the specific electronic dynamics of the bands (linear dispersion near the *K* points for electrons near the Fermi energy ($E_F$), trigonal warping for hot electrons, etc.) into the WPD calculation and (ii) it allows us to handle localized defects. We were able to exploit this feature of the pseudopotential in calculating the transport properties of different 0D and 1D graphene defects [22,23].

The WPD calculation has two input quantities: (i) the Hamiltonian and (ii) the $\psi_0\left(\vec{r}\right) = \psi\left(\vec{r}, t = 0\right)$ initial state. In our former calculations $\psi_0\left(\vec{r}\right)$ was a simple 3D Gaussian wave packet (WP), but this made it necessary to start the initial WP from a region of space, where $V\left(\vec{r}\right) = 0$, or at least constant, and the WP had to be injected from this region into the nanosystem through a simulated interface layer. This, however, made it difficult to create a WP with pre-determined initial spectrum on the nanosystem, because the interface layer can considerably distort the spectrum, which can also affect the scattering processes and thus the final state strongly depends on the way the initial state was prepared. For example, when the initial WP is injected on the graphene sheet from a simulated STM tip, a complicated multiple scattering process occurs between the tip and the graphene surface, which selects only certain components. The further spreading on the graphene surface will be determined by the initial state formed by these specific components [18]. Such processes have a profound influence on the STM imaging, resulting in anisotropic currents even on the Fermi energy.

In this paper, to avoid this problem, a tailor-made initial WP on the nanosystem is built by constructing the WP as a superposition of quasiparticle Bloch states, which is presented in Section 2. These precisely determined WPs are used to study the scattering on defect states. In Section 3, we introduce and apply an alternative WPD technique, which has the advantage that it completely circumvents the need to calculate a pseudopotential. We work out the time evolution of the WP directly from the $E\left(\vec{k}_{Bloch}\right)$ dispersion relation. Formerly, we applied this method for studying the WPD on pristine graphene and $MoS_2$ single layers [20] and now we extend it for graphene structural defects opening the way for the WPD investigations of defect scattering processes in other 2D materials.

## 2. Bloch Function Wave Packet Construction, Time Evolution and Scattering in Graphene

For any $V\left(\vec{r}\right)$ potential which is periodic in space, the solutions of the stationary Schrödinger equation have the form of $\varphi\left(\vec{r}, \vec{k}_{Bloch}\right) = u\left(\vec{r}, \vec{k}_{Bloch}\right) e^{i \vec{k}_{Bloch} \vec{r}}$, where $u\left(\vec{r}, \vec{k}_{Bloch}\right)$ is a periodic function and $\vec{k}_{Bloch}$ is the Bloch wave vector. These $\varphi\left(\vec{r}, \vec{k}_{Bloch}\right)$ functions are called Bloch functions. Bloch functions and the corresponding $E\left(\vec{k}_{Bloch}\right)$ energies can be easily computed numerically for any periodic potential, with the Fourier transformed form of the Schrödinger equation. Figure 1 shows some characteristic Bloch functions for the graphene surface, computed by using the graphene pseudopotential [21].

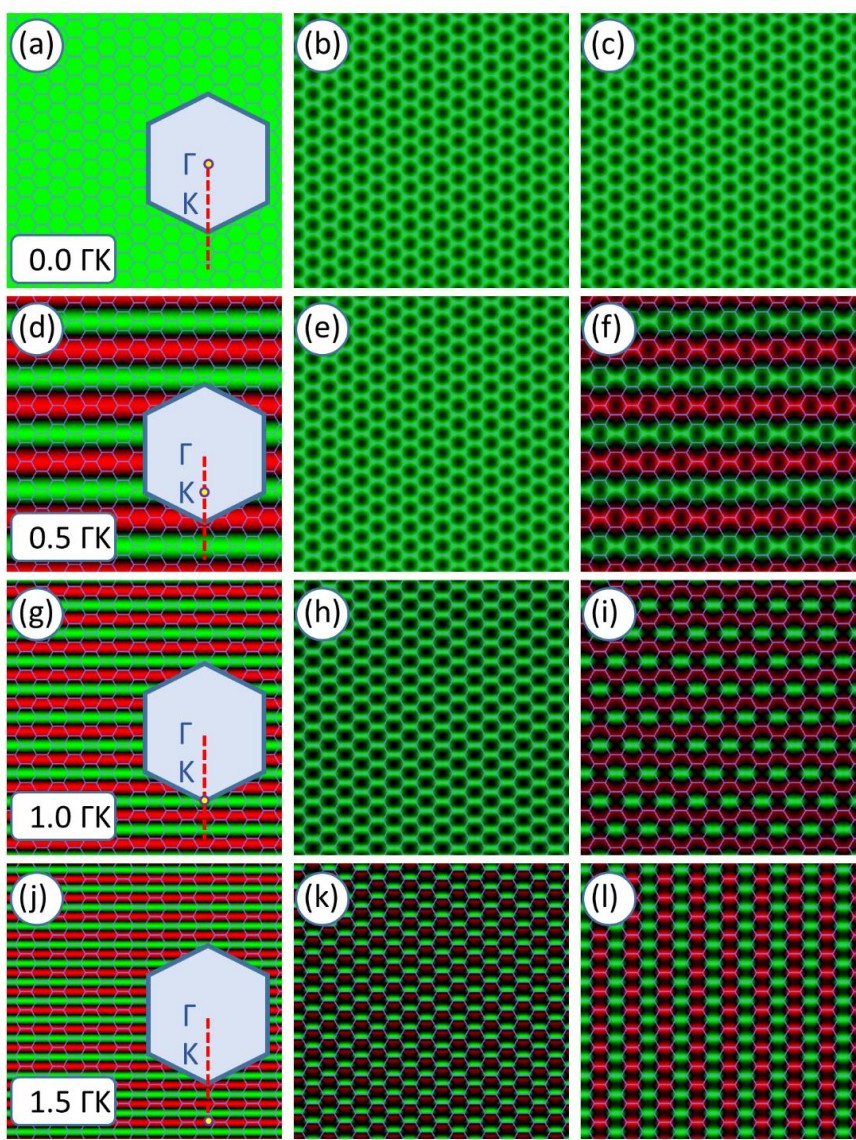

**Figure 1.** Construction of the graphene Bloch wave functions for different points along the ΓK line in the extended Brillouin zone. Left column: $e^{i \vec{k}_{Bloch} \vec{r}}$. Middle column: $u\left(\vec{r}, \vec{k}_{Bloch}\right)$. Right column: $\varphi\left(\vec{r}, \vec{k}_{Bloch}\right)$. See Section 2 for details. Real parts of the wave functions are shown, green is positive, red is negative. The graphene lattice is shown by blue lines. The insets in a, d, g, j show the position of $\vec{k}_{Bloch}$ (yellow dot) relative to the Brillouin zone.

The symmetry properties of Bloch functions depend both on the symmetry of the lattice and the $\vec{k}_{Bloch}$ Bloch wave vector. The effect of $\vec{k}_{Bloch}$ on the Bloch function is twofold: (i) the $e^{i\vec{k}_{Bloch}\vec{r}}$ plane wave part introduces a $\vec{k}_{Bloch}$ dependent phase factor and (ii) the shape of the $u\left(\vec{r},\vec{k}_{Bloch}\right)$ function depends on $\vec{k}_{Bloch}$. The $\vec{k}_{Bloch}$ dependent phase factors determine the interference patterns of two or more Bloch functions. Such interference patterns are visible in FFT-LDOS images obtained by STM. In the case of the Γ point (Figure 1a–c), the symmetry of the wave function is identical to the potential itself, a hexagonal symmetry. However, increasing the Bloch wavelength along the vertical direction (Figure 1d–l) introduces a *y* dependent modulation into the Bloch function.

Next, we constructed localized wave packets as a superposition of Bloch states. It is possible to build maximally localized Wannier functions [24,25] by a carefully chosen superposition, but those superpositions span in the whole Brillouin zone. If we want to have a fairly narrow spectral distribution together with a sufficiently narrow spatial distribution, we can use a superposition with a simple Gaussian amplitude function $a\left(\vec{k}\right)$:

$$\phi\left(\vec{r}\right) = \int e^{-\frac{\left|\vec{k}-\vec{k}_0\right|^2}{4\Delta k^2}} e^{i\vec{r}_0\vec{k}}\, \varphi\left(\vec{r},\vec{k}\right) d^3\vec{k}, \tag{1}$$

where we wrote $\vec{k}$ for $\vec{k}_{Bloch}$ in this formula for brevity (see Appendix A), $\vec{k}_0$ is the momentum space vector, $\vec{r}_0$ is the direct space initial position of the Gaussian and $\Delta k$ is its momentum width.

Figure 2a shows the constructed WP localized in its *y* coordinate in direct space. Its spectrum is concentrated on one of the graphene *K* points and Figure 2g shows the plot of the real part of the $a\left(\vec{k}\right)$ amplitude function, which is a Gaussian multiplied with a plane wave. Figure 2a–c show the time evolution of this Bloch WP for an unperturbed graphene lattice calculated by numerically solving the time-dependent Schrödinger equation with the split-operator FFT method [26–28]. We used the $\hat{H} = \hat{K}_{free} + \hat{V}$ Hamilton operator, where $\hat{K}_{free}$ is the free space kinetic energy operator and $\hat{V} = V\left(\vec{r}\right)$ is the graphene pseudopotential [21]. This time evolution—i.e., that for the infinite lattice—can also be analytically calculated by inserting the $exp\left[-iE\left(\vec{k}_{Bloch}\right)t\right]$ time propagator into the kernel of Equation (1), where $E\left(\vec{k}_{Bloch}\right)$ is the graphene dispersion relation. The WP moves in the $-y$ direction, because its spectral distribution is centered at the lowest *K* point of the graphene Brillouin zone. The spectral width of the WP is small, $0.1\overline{\Gamma K}$, hence the dispersion relation is still in the linear regime. There is only a very small spreading in the time evolution resulting in that the width and the shape of the WP remain unchanged.

Figure 2d–f show the time evolution of the same initial Bloch WP on a graphene lattice with a structural defect. The defect was modeled by adding a Gaussian potential to the pseudopotential. The Gaussian was centered on an atomic site, its height is 54 eV and its HWHM is 0.45 nm. This raises the $-44$ eV minimum and $+21$ eV maximum values of the pseudopotential to $+11$ eV and $+75$ eV, respectively. These parameters were chosen in order to demonstrate the anisotropic spreading effects, see below. In the future, we intend to calculate defect potentials that model real structural defects. We can see that the WP is indeed scattered on the defect and the angular distribution of the scattered WP has a hexagonal symmetry (Figure 2h and Video S1). Similar anisotropic property around localized defect was also proved by using analytical calculations of the Green's function [29] highlighting the accuracy of our WPD method. It is worth noting that anisotropy was also observed in the experiments around a point defect [30] originating from intervalley scattering processes. In order to directly reveal the intervalley scattering process in our

WPD simulations we performed Fourier transformation on our calculated probability density images (Figure 2) and compared the FFT images with our STM measurements.

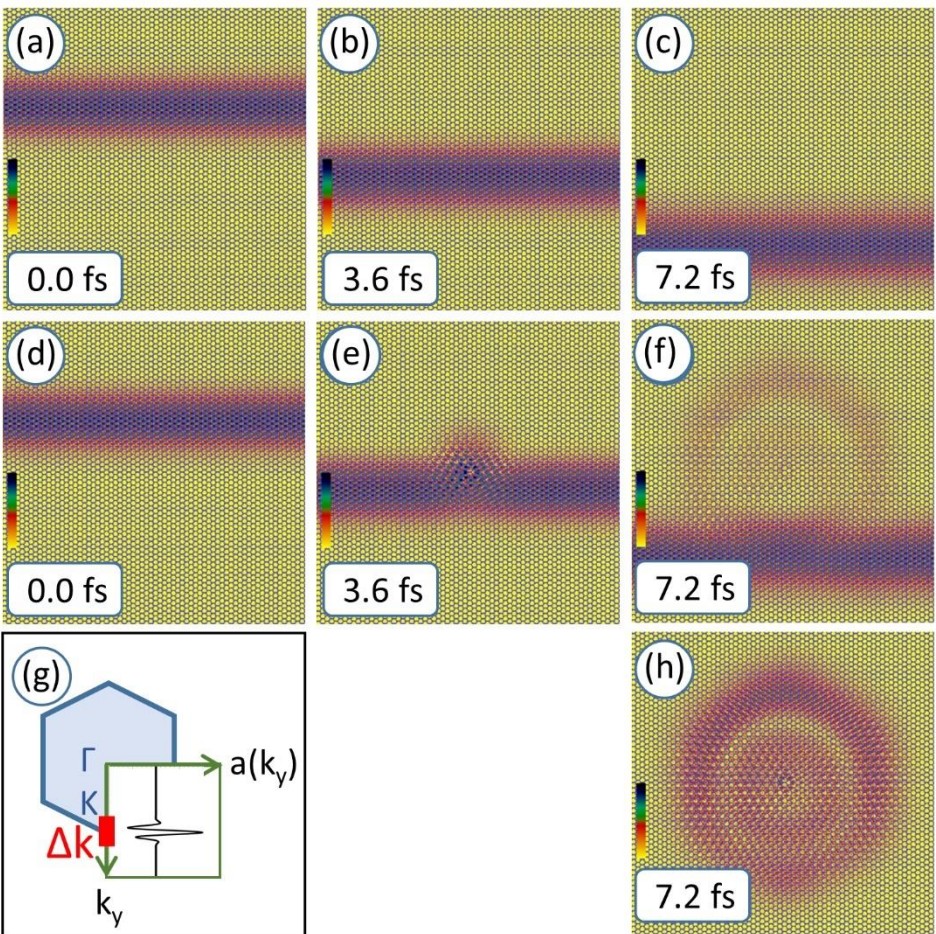

**Figure 2.** Time evolution of the probability density of a Bloch function wave packet in the graphene pseudopotential. (**a**–**c**) Without defect. (**d**–**f**) With defect. (**g**) Spectral distribution of the Bloch function wave packet in the Brillouin zone, only the real part of $a\left(\vec{k}\right)$ is shown. The red bar shows the reciprocal space width $\Delta k$. (**h**) Scattering pattern, absolute value of the difference of the WPs shown on (**c**,**f**). The size of the calculation window is 14.48 nm. (see Video S1 for the time development of the Bloch WP).

Figure 3a is the 2D-FFT power spectrum of the probability density of the initial wave packet, $F\left[|\psi(x,y;t=0)|^2\right]$. This Fourier image shows a thin vertical line section in the origin, repeated six times at the vortices of a hexagon. The line section has Gaussian density distribution along $y$. It is the reciprocal space representation of the initial wave packet (Figure 2d), which is delocalized in the x direction and localized in the y direction. Figure 3b is the 2D-FFT power spectrum of the probability density of the scattered wave packet, $F\left[|\psi(x,y;t=7.2fs)|^2\right]$. The $t = 7.2fs$ corresponds to a specific time value, when the WP has already scattered on the defect (Figure 2f), therefore, the incoming and scattered waves can lead to characteristic interference patterns, which can be compared with the FFT images of the STM measurements. In Figure 3b, we can still recognize the spot at the origin and its six copies at the vortices of the hexagon, but the spots are not narrow vertical lines anymore. They have indeed a finite size in both directions, because the scattering is caused by the localized defect: the incoming wave packet has only vertical momentum, but the outgoing wave packet has momentum components distributed along the whole polar angle. Six other spots (marked by green circles) are also visible on this Fourier image. These

are the characteristic signatures of the intervalley scattering often seen on experimental FFT-LDOS images. For comparison, Figure 3c shows a typical 2D FFT-LDOS of a measured topographic STM image around the defect, where intervalley scattering occurs. The STM image was taken on a HOPG graphite sample, near a monoatomic step. The tunnel current was 1 nA and the STM bias was 100 mV.

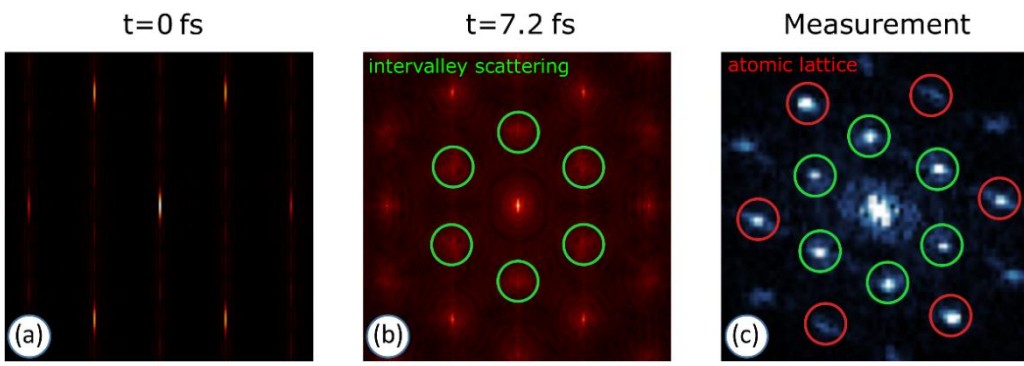

**Figure 3.** Simulated and measured FFT-LDOS maps. (**a**) Two-dimensional Fourier transform (2D FFT) of the probability density of the initial wave packet shown on Figure 2d. (**b**) 2D-FFT of the probability density at t = 7.2 fs (Figure 2f). (**c**) 2D-FFT of a topographic STM image.

### 3. Band Structure Governed Wave Packet Dynamics

In the traditional formulation of the WPD, the environment of the moving electron quasiparticle (i.e., the crystal and the averaged effect of the other electrons) is incorporated into the potential energy operator, often by the help of a (pseudo)potential, thus the Hamiltonian is $\hat{H} = \hat{K}_{free} + \hat{V}$, where $\hat{K}_{free}$ is the kinetic energy operator for a free electron. In this section, we present an alternative formulation, where the effect of the crystal potential and the many-body effects are entirely incorporated into the kinetic energy operator, i.e., the Hamiltonian is $\hat{H} = \hat{K} + \hat{V}_{free}$.

In our alternative formulation all crystal structure and electronic structure effects are taken care of in band structure calculations that yield a dispersion relation $E_n\left(\vec{k}_{Bloch}\right)$, where $\vec{k}_{Bloch}$ is the Bloch wave vector and $n$ is the band index. Thus, we replace the kinetic energy operator in free space, $\hat{K}_{free} = \dfrac{\left|\vec{k}\right|^2}{2} = E_{free}\left(\vec{k}\right)$ in momentum representation, where $E_{free}\left(\vec{k}\right)$ is the free space dispersion relation, with the kinetic energy operator that describes the dispersion of a many-electron system in a given crystalline material. Thus, the momentum representation of the kinetic energy operator becomes $\hat{K} = E\left(\vec{k}_{Bloch}\right)$.

By using this modified kinetic operator $\hat{K} = E\left(\vec{k}_{Bloch}\right)$, a perfect crystal Hamiltonian (without any defect) can be written as $\hat{H} = \hat{K} + \hat{V}_{free}$, where $\hat{V}_{free} = 0$ by definition. We have already presented such calculations for perfect 2D crystals, graphene and $MoS_2$ single sheets in Reference [20]. Those calculations successfully reproduced the trigonal warping effect and the anisotropic WP spreading characteristic of these 2D materials and showed different symmetries of the WP spreading, depending on the band structure and the spectral distribution of the initial WP. As we emphasized in Reference [20], similar calculations can be easily performed for any crystalline material, where the dispersion relation is known.

In this section, we combine the two methods to study defects in 2D crystalline materials. We describe the (infinite, periodic) crystal by the kinetic energy operator (utilizing the dispersion relation of the material) and the structural defect by a local potential. Thus,

our Hamiltonian will become $\hat{H} = \hat{K}_{crystal} + \hat{V}_{defect}$, where the $\hat{K}_{crystal} = E\left(\vec{k}_{Bloch}\right)$ kinetic energy operator describes the electronic structure of the crystal lattice and the $\hat{V}_{defect} = V\left(\vec{r}\right)$ potential energy operator describes the local defect. The main novelty of this combined method is the straightforward treatment of the different types of defects in several 2D materials.

Figure 4a–c show the time evolution of a WP for an unperturbed graphene lattice and Figure 4d–f show the time evolution in the presence of a local defect calculated by numerically solving the time-dependent Schrödinger equation. The defect was modeled by a Gaussian potential, its height is 2 eV and its HWHM is 0.8 nm. In contrast to the previous Section, now we used a localized initial state. The spectral distribution of the initial WP is a sum of six Gaussians, placed in the six $K$ and $K'$ points (Figure 4g) resulting in a spatially well-localized initial state. Such an initial state models a situation, where the initial WP is injected onto the graphene sheet from a localized source.

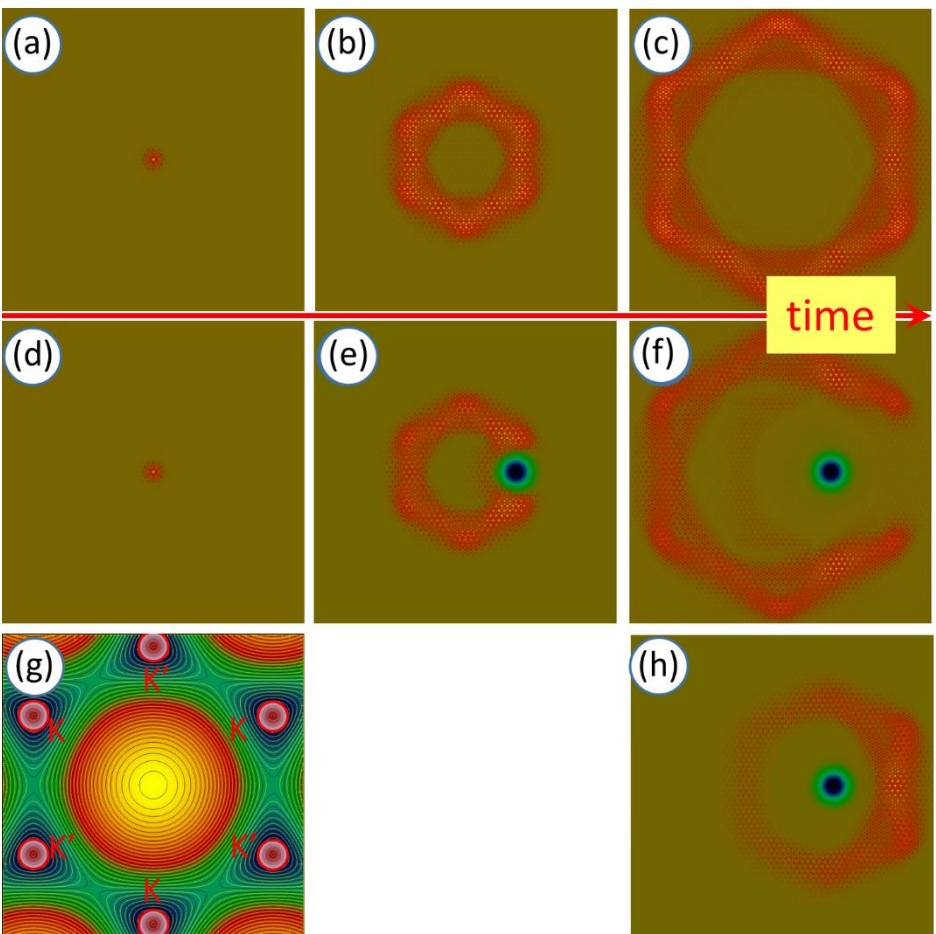

**Figure 4.** Time evolution of the probability density of a wave packet on the graphene surface with- and without defect. The band structure of the perfect crystal is incorporated into the kinetic energy operator, hence the potential is everywhere zero, except in the defect region. (**a–c**) Without defect. (**d–f**) With defect. The defect is shown by a blue spot. (**g**) Spectral distribution of the initial wave packet in the Brillouin zone (absolute value) is shown by red circles in the $K$ and $K'$ points, superimposed on the graphene band structure. The radius of the red circles is $\Delta k/2$, the intensity of the red color is proportional to the spectral weight at each $\left(\vec{k}_x, \vec{k}_y\right)$ points. (**h**) Scattering pattern, absolute value of the difference of the WPs shown on (**c,f**). The size of the calculation window is 23.04 nm.

Several important effects can be seen in the time evolution of the probability density of the WP. Firstly, in the case of the unperturbed graphene surface, we can see an anisotropic (hexagonal) spreading of the WP. This effect is related to the spatially localized initial wave function and is in good agreement with our earlier calculations [18], where the initial WP was injected onto the pristine graphene surface from an STM tip. More recently, this effect was studied also with TB-DFT calculations, where a model gold STM tip was used [31], as well as with Green's function formalism in a dual-probe STM setup [32]. In each case, long-range oscillation occurs resulting in an increased conductance in the armchair directions of the pristine material due to the localized source of the electrons. Secondly, an atomic scale modulation is presented on the WP. This is surprising at first glance, because this calculation does not directly have the atomic lattice as input. The atomic structure is, however, implicitly represented by the graphene dispersion relation. Thirdly, when the WP hits the local potential, the scattered WP has a hexagonal symmetry (anisotropic)—as it is best seen on the difference image in Figure 4h, though the local potential is cylindrically symmetric (isotropic). This effect, which can be termed as a "generalized Huygens' principle", is caused by the anisotropic nature of the graphene dispersion relation. Huygens' principle states that all points of a wave front may be regarded as new sources of wavelets that expand in every direction at a rate that depends on their velocities [33]. In the case of an isotropic medium, the wavelets have a spherical shape (circles in 2D). If the value of these velocities, however, depends on the angle of propagation, then the wave front does not remain circular, and we obtain an anisotropic wave propagation [34].

## 4. Conclusions

We presented an extension to the wave packet dynamical method, where the initial wave packet is constructed from the numerically calculated quasiparticle Bloch states of the pseudopotential that represents the physical system. This method makes it possible to fine-tune the spectral distribution of the initial wave packet and thus to investigate the details of the scattering processes and the nanoscale transport phenomena for different scenarios. The method was applied for a structural defect on a graphene surface, where we studied the scattering of the Bloch function wave packet on the local defect. A hexagonal scattering pattern was seen associated with intervalley scattering. Then we calculated the wave packet time evolution by a second method, where the properties of the infinite crystal (its atomic- and electronic structure) are coded into the kinetic energy operator and the properties of the local defect are represented by a local potential. We show that the hexagonal spreading pattern is already present for the pristine graphene, where the initial wave packet is spatially well localized. This initial WP represents a physical system, where the electrons are injected locally into the graphene surface as in the case of an STM imaging process. We found an anisotropic (hexagonal) scattering pattern emerging even for an isotropic potential, similar to the applied pseudopotential case. The advantage of the method is that it does not require the calculation of a pseudopotential. The only input necessary to represent the infinite crystal is its $E\left(\vec{k}_{Bloch}\right)$ band structure, available from, for example, a simple DFT calculation.

Our extended wave packet dynamical method clearly demonstrated the possibilities to investigate defects in different 2D materials, which serve as a new platform to better understand the basic mechanism of quasiparticle scattering processes at the atomic scale. We intend to apply this method for TMDC materials [35–37], where the strong spin-orbit coupling leads to large spin splitting with opposite spin directions of the *K* and *K'* valleys, further modifying the possible scattering processes. Another possible application of our method is related to twisted bilayers [38,39], where symmetry-breaking correlation effects might take place, which have important consequences on the QPI patterns. In all cases, the scattering processes calculated by the wave packet dynamical method and their comparison with the measured QPI patterns by using STM will provide useful information to the application of 2D materials in nanodevices.

**Supplementary Materials:** The following are available online at https://www.mdpi.com/article/10.3390/app11114730/s1, Video S1: time development of a Bloch function wave packet on the graphene surface.

**Author Contributions:** Conceptualization, G.I.M. and P.V., methodology, software, G.I.M.; STM experiment, P.N.-I., writing—original draft preparation, G.I.M., A.M. and P.V. All authors have read and agreed to the published version of the manuscript.

**Funding:** The work was supported by the European H2020 GrapheneCore3 Project No. 881603, Graphene Flagship, the NanoFab2D ERC Starting Grant and the Hungarian National Research, Development and Innovation Office (Grant No. KH130413). P.V. acknowledges the support of the Janos Bolyai Research Scholarship of the Hungarian Academy of Sciences. A.M. is funded by the Fund for Scientific Research (F.R.S.-FNRS) of Belgium. P.N.-I. acknowledges support from the Hungarian Academy of Sciences, Lendulet Program, Grant No. LP2017-9/201.

**Conflicts of Interest:** The authors declare no conflict of interest.

**Appendix A**

In this paper, the kinetic energy operator is defined in momentum space and the potential energy operator is defined in direct space. This is well suited to the calculation method, because we solve the time-dependent Schrödinger equation by the split-time FFT method, where in each time step $\Delta t$ we change the wave function from the direct to the reciprocal space and back by Fourier transforms. We have to emphasize that the term "Fourier transform" is used in two distinct meanings in this context. The split-time FFT method uses the fast Fourier transform (FFT), which decomposes the wave function into plane waves, represented by $\vec{k}$ wave vectors. In a crystalline material, described by a periodic potential, $\vec{k}$ is not a good quantum number, but the crystal momentum $\vec{k}_{Bloch}$ is. The momentum wave function $\varphi\left(\vec{k}_{Bloch}\right)$ can be computed from the direct space wave function $\varphi\left(\vec{r}\right)$ by applying a generalized Fourier transform, where the basis functions are the Bloch waves corresponding to the $V\left(\vec{r}\right)$ potential.

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
