# Peer review of "Wave Packet Dynamical Simulation of Quasiparticle Interferences in 2D Materials†"

_applsci, doi:10.3390/app11114730_

Round 1

Reviewer 1 Report

The manuscript under consideration is original and will be interesting for the readers. The quality of the paper preparation is good enough. I didn't find any serious flaws. So I thing that the paper may be published in its present form without any unnecessary delay.

Author Response

We would like to thank the Referee for finding our work interesting.

Reviewer 2 Report

Nowadays, two-dimensional materials are attracting increasing interest due to their unique physical characteristics. However, their functional properties are greatly influenced by structural defects. Vasco et al. presented two WPD-based calculation methods for a graphene sheet with a localized defect. In both cases calculation have been carried out for an ideal crystal lattice and a defect one. The authors present the results of their study in an accessible and clear way. The object of the study fully corresponds to the aims and scopes of the journal. The manuscript can be accepted for publication, although minor corrections are necessary.

Line 47, "scanning tunneling spectroscopic microscopy": it would be better if the authors used one word "spectroscopy" or "microscopy". Since the following sentence uses the acronym STM, it would be "scanning tunneling microscopy”.

Figure 1. In the figure caption the authors refer to section 1, “Introduction”, for details. I recommend adding more detailed related comments to the figure in the text of section 2 or in the caption to the figure for better understanding by readers.

Author Response

We would like to thank the reviewer for the constructive evaluation of the paper. We have added further information to the text related to Figure 1. We have also revised the English of the paper.

Reviewer 3 Report

In this paper the authors propose  new calculation methods to describe the scattering of a quasi particle WPs on localised defects on a graphene sheet. The authors also describe the  scattering time evolution by the time development of the WPs. The paper is well written, and their calculations are validated by the experimental data.  In my opinion the authors should explain better how they simulate the time scattering evolution  reported in Fig. 3. What does the value of 7.2fs  represent? can the authors be more clear on that?

In my opinion the paper can be pubished after these minor revisions.

Author Response

We would like to thank the Referee for finding our work interesting.

The t=7.2 fs in the FFT-LDOS map (Figure 3b) corresponds to a specific time value, when the wavepacket has already scattered on the defect, therefore the incoming and scattered waves can lead to characteristic interference patterns (see Figure 2f). In this case, two-dimensional Fourier transformation of this probability density image can be compared with the STM experiments (Figure 3c), where the same standing wave patterns are measured in the Fourier transformed images. We note that this time value (t=7.2 fs) can be calculated in a simple way considering the initial position of the wavepacket measured from the defect site and the Fermi-velocity of the wavepacket.

The manuscript text has been amended with additional information related to the scattering results of Figure 3.